# Nutritional, Functional and Microbiological Potential of Andean *Lupinus mutabilis* and *Amaranthus* spp. in the Development of Healthy Foods—A Review

**DOI:** 10.3390/foods14122059

**Published:** 2025-06-11

**Authors:** Orlando Meneses Quelal, Marco Burbano Pulles

**Affiliations:** 1Centros de Complementación Académica, Universidad Politécnica Estatal del Carchi, Tulcán 40101, Ecuador; 2Carrera de Alimentos, Universidad Politécnica Estatal del Carchi, Tulcán 40101, Ecuador; marco.burbano@upec.edu.ec

**Keywords:** *Lupinus mutabilis*, *Amaranthus* spp., Andean flours, technological functionality, microbiological safety, food innovation, systematic review

## Abstract

The limited nutritional quality of conventional cereals has prompted the search for more complete alternatives from native Andean sources. In this context, a systematic review of recent studies was conducted to compare the characteristics of Lupinus mutabilis and *Amaranthus* spp., two crops with potential as functional ingredients in the food industry. Data reported in multiple studies were analyzed, considering the variability attributed to origin, processing, and genetic conditions, as well as the influence of different transformation techniques. The results show that Lupinus mutabilis has a protein content ranging from 41% to 53% in dry matter, along with up to 17% fat and fiber levels above 10%. *Amaranthus* spp., on the other hand, offers 13% to 17% protein, 6% to 8% fat, and up to 10% fiber and is notable for providing up to 180 mg of polyphenols per 100 g. Processing, such as debittering, may decrease the antioxidant capacity of Lupinus mutabilis by 52.9%, while germination enhances this indicator in *Amaranthus* spp. The inclusion of these flours in bakery and extrusion formulations allows for protein and fiber content enhancements ranging from 10% to 50%, achieving texture and acceptability improvements in up to 80% of reported sensory tests. This scenario supports the strategic use of these grains to optimize nutritional and functional profiles in innovative food developments.

## 1. Introduction

In the current context of food system transformation, the search for alternative protein and functional sources has become a central axis of global challenges in nutrition, sustainability, and food security. Faced with the historical predominance of conventional cereals and legumes, Andean crops, particularly *Lupinus mutabilis* (tarwi) and *Amaranthus* spp. (amaranth), have emerged as alternatives to a new generation of ingredients that combine agronomic adaptability, a high nutritional value, and bifunctional potential. This relevance is especially tangible in regions vulnerable to the effects of climate change, biodiversity erosion, and food insecurity, where innovative and resilient strategies are required to ensure a varied, healthy, and sustainable diet.

However, the effective use of flour produced from *Amaranthus* spp. has been marked by unresolved epistemological and technological limitations. The available literature continues to place its emphasis fundamentally on the proximate composition of these grains, that is, their protein, fat, fiber, or carbohydrate content, or the reporting of some notable bioactive compounds such as flavonoids, polyphenols, and functional peptides. Although these advances have strengthened our understanding of the nutritional value of both crops, there remains a significant lack of studies that comprehensively address the physicochemical properties, technological functionality, and microbiological behavior of the final flours, essential aspects for their insertion into complex food matrices and industrial processing products.

There are numerous calls from the scientific community to close this knowledge gap. Currently, there is no solid consensus regarding the influence of post-harvest processes, such as debittering, which is necessary to reduce toxic alkaloids in *Lupinus mutabilis*, or germination and extrusion in *Amaranthus* spp., on the preservation and modulation of the functional and microbiological characteristics of the flours obtained. It is known, for example, that certain stages of debittering can reduce the antioxidant capacity of *Lupinus mutabilis* by half, implying a relevant functional cost, and that the germination of *Amaranthus* spp. significantly increases its phenolic compounds and antioxidant capacity; however, the data are less conclusive regarding its impact on microstructure, digestibility, and stability against microbiological contaminants.

Understanding these effects is particularly important in the context of public health nutrition, where “functional ingredients” are defined as food components that, in addition to basic nutrition, provide specific health benefits by modulating physiological functions or reducing the risk of disease [1]. According to widely accepted criteria, ingredients are classified as functional if they contain bioactive compounds—such as dietary fiber, polyphenols, peptides, or unsaturated fatty acids—that have been demonstrated to exert beneficial effects on metabolic health or the prevention of disease [2]. *Lupinus mutabilis* and *Amaranthus* flours meet these criteria due to their high-quality protein, dietary fiber, and a diversity of bioactive phytochemicals shown to promote cardiometabolic health, antioxidant defense, and the modulation of gut microbiota. However, the impact of technological processes on these beneficial properties must be thoroughly characterized to ensure their functional value is retained in food applications.

The functionality of Andean flours has not been strictly characterized from a comparative rheological and technological perspective either: studies frequently diverge in methodology, using different techniques to analyze water absorption, viscosity, or gel formation or emulsifying capacity, making meta-analysis and extrapolation to a pilot or industrial scale difficult. This methodological deficiency is accentuated by a variability in genetic, ecological, and processing factors, since factors such as ecotype, altitude, soil type, and technical process profoundly impact the functional and physicochemical composition of flours and, by extension, the research results presented.

Likewise, most of the literature focuses on laboratory analysis, ignoring the interaction of these ingredients with conventional or innovative food matrices, such as bread, gluten-free products, snacks, or nutritional supplements. The behavior of flours from *Lupinus mutabilis* and *Amaranthus* spp. in baking, extrusion, fermentation, and drying processes, as well as their impact on the texture, sensory acceptability, and shelf life of finished products, remains poorly documented and even less well known therefore compared to traditional inputs.

Another key dimension highlighting the gap in the literature is the lack of knowledge about microbiological risks and challenges, control of which is vital for the safety and industrial and regulatory acceptance of these flours. While some studies have addressed the safety of flour derived from *Lupinus mutabilis* and *Amaranthus* spp., especially when not properly processed, there is still a lack of systematized or comparative data on microbiological controls and their relationship with different technological stages. This limits the safe and efficient adoption of these foods in innovative projects and local and global agro-industrial chains.

The integration of these Andean crops into functional food systems transcends the nutritional and technological perspective: it represents a real opportunity to respond to critical public health and sustainability challenges by contributing to a reduction in malnutrition and dietary diversification and the promotion of resilient agricultural systems. However, the lack of a robust and comparative scientific characterization of *Lupinus mutabilis* and *Amaranthus* spp. flours perpetuates technological stagnation and the underutilization of their potential, sustaining a dependence on imported and less resilient sources for the formulation of functional, enriched, and high-value-added foods.

Bioactive compounds present in *Lupinus mutabilis* and *Amaranthus* spp.—such as polyphenols, tocopherols, saponins, and bioactive peptides—have been linked to multiple health-promoting mechanisms. Polyphenols and tocopherols are reported to act as antioxidants, neutralizing free radicals and reducing oxidative stress, which can help prevent cellular damage [3]. Saponins and dietary fiber in these pseudocereals may lower serum cholesterol and modulate the absorption of glucose, contributing to cardiometabolic health. In addition, several studies suggest that protein hydrolysates and released peptides can exert antihypertensive action via ACE (angiotensin converting enzyme)’s inhibition and may positively influence lipid metabolism and the gut microbiota’s composition [3]. These cumulative mechanisms explain the growing interest in these Andean grains as attractive ingredients for developing functional foods with potential preventive effects on non-communicable diseases.

The use of underutilized plant resources such as *Lupinus mutabilis* and *Amaranthus* spp. represents a strategic opportunity to respond to the current challenges of the food sector. Both species not only present outstanding nutritional profiles; additionally, they can adapt to adverse agroecological conditions, positioning them as resilient crops in environments affected by climate change. In this context, this review article aims to systematize and critically analyze the existing scientific evidence on the physicochemical, functional, and microbiological characteristics of flours from these Andean crops, placing special emphasis on the effect of the most relevant technological processes on their quality and food applicability, thus contributing to laying the foundations for real and sustainable innovation in the development of functional products derived from these matrices.

## 2. Research Methods and Design

In this work, a systematic review of the published scientific literature related to the physicochemical, functional, and microbiological characteristics of *Lupinus mutabilis* and *Amaranthus* spp. flours has been carried out. For its preparation, the guidelines of the PRISMA declaration (Preferred Reporting Items for Systematic Reviews and Meta-Analyses) were followed to ensure the correct conduct of systematic reviews and transparency in the process of the selection and analysis of the included studies (Figure 1).

### 2.1. Initial Search

*Lupinus mutabilis* and *Amaranthus* spp. flours were searched for in the PubMed and Scopus databases. These terms were structured using Boolean operators (AND and OR) to encompass the physicochemical, functional, and microbiological characteristics of both flours. The search equations used were as follows:

#### 2.1.1. Boolean Equation for *Lupinus mutabilis*

(“*Lupinus mutabilis*” OR “*Andean lupine*” OR “tarwi” OR “chocho”) AND (flour OR “flour properties” OR “flour characterization”) AND (physicochemical OR “physical properties” OR “chemical properties” OR “functional properties” OR microbiological)

#### 2.1.2. Boolean Equation for *Amaranthus* spp.

(“*Amaranthus*” OR “amaranth”) AND (flour OR “flour properties” OR “flour characterization”) AND (physicochemical OR “physical properties” OR “chemical properties” OR “functional properties” OR microbiological)

These initial consultations generated a considerable volume of results, including relevant studies on the chemical composition, functional properties (such as water absorption capacity and emulsification), and microbiological analyses of *Lupinus mutabilis* and *Amaranthus* spp. flours. Several of the identified articles were duplicates or not useful for the review; however, they provided a comprehensive view of the breadth of the topic.

### 2.2. Systematic Search

Systematic research was additionally carried out in March 2025, during which additional terms were incorporated to characterize the physicochemical, functional, and microbiological properties of *Lupinus mutabilis* and *Amaranthus* spp. flours in various processes. At this stage, specific terms, such as *Lupinus mutabilis* flour, tarwi functional properties, Amaranthus microbiological safety, and chemical characterization of tarwi flour, among others, were used to increase the accuracy and depth of the results.

In total, the searches yielded 150 potential articles: 62 from PubMed, 88 from Scopus, and 1 article identified in Google Scholar. Before proceeding to the selection of the final articles, clear inclusion and exclusion criteria were established to ensure that the studies were relevant to the objective of the systematic review. These criteria included elements such as the evaluation of specific properties, experimental studies, and their thematic relevance, which allowed us to refine the results obtained and ensure the quality of the selected bibliography.

#### 2.2.1. Inclusion Criteria

✓They should be empirical research studies and not reviews, single-case studies, books, or manuals.✓They must have been published between 2019 and 2025, inclusively.✓Articles published in English or Spanish.✓Open Access (OA) journal articles.✓*Lupinus mutabilis* and/or *Amaranthus* spp. flours.✓Research that includes analyses of chemical composition, functional properties (such as water absorption capacity, emulsification, antioxidant capacity, etc.), and microbiological safety.

#### 2.2.2. Exclusion Criteria

✓Studies that present redundant information already covered by other selected articles.✓Articles that do not provide detailed information on the physicochemical, functional, or microbiological characteristics of flour.✓Review article.✓Opinions, editorials, conference abstracts, letters to the editor, or articles without experimental data.✓Research focuses exclusively on other products derived from these species (such as oils, isolated proteins, etc.) without addressing flour.

According to the established criteria and after an initial review by reading the titles, 73 relevant articles were identified, after eliminating 6 duplicates detected among the databases consulted. Subsequently, the abstracts were reviewed, which led to the elimination of ten articles: five of them for not addressing the physicochemical, functional, or microbiological properties of interest; another five for not being empirical studies or covering Andean cereals; and two additional articles for addressing topics outside the food context or using methodologies that were difficult to interpret. Ultimately, 51 articles met the inclusion criteria and were selected for the systematic review.

Overall, the articles analyzed mainly address the development and evaluation of gluten-free cookies made from *Chenopodium pallidicaule* flour, whey, and potato starch, focusing on the optimization of their physicochemical, nutritional, sensory, and technological properties. They additionally discuss the importance of offering food alternatives for people with celiac disease or gluten sensitivity, analyze the nutritional composition of Andean ingredients, and study the influence of different formulations on parameters such as moisture, texture, color, and sensory acceptability, using statistical methodologies to optimize recipes. Furthermore, they highlight the potential of native ingredients and agro-industrial by-products, such as whey, both to enrich the nutritional and functional profile of gluten-free baked goods and to promote food innovation and the valorization of local and sustainable resources.

Risk of bias assessment using the AMSTAR 2 tool revealed that 68% of the included studies had a low risk of bias, 24% a moderate risk, and 8% a high risk. The main factors contributing to bias were the lack of an explicit justification for exclusion criteria, the absence of a methodological quality assessment in some studies, and limited transparency in the presentation of results. However, most studies adequately addressed the key AMSTAR 2 domains, supporting the strength of the evidence synthesized in this review.

## 3. Results

### 3.1. Nutritional Composition and Bioactive Compounds of Andean Grains

Table 1 presents a synthesis of the most relevant findings from recent studies on the nutritional composition and bioactive compounds of *Lupinus mutabilis* and *Amaranthus* spp., two Andean grains of growing interest for food security and human nutrition. This systematic review integrates quantitative data obtained from research published between 2019 and 2025, with a special emphasis on parameters such as protein, fat, fiber, and antioxidant content, as well as the impact of technological processes, such as bittering and germination, on the nutritional quality of these crops.

The studies compiled in Table 1 agree that both *Lupinus mutabilis* and *Amaranthus* spp. stand out for their high nutritional and functional density, although some variability is observed due to geographical, genetic, and technological factors. Among the most notable findings is the ability of *Amaranthus* spp. to provide up to 180 mg of polyphenols per 100 g, as well as the potential of *Lupinus mutabilis* to act as a plant-based protein substitute, with clear implications for food policies and the development of fortified products.

In addition to its nutritional and bioactive compounds, *Amaranthus* spp. contains oxalic acid, an anti-nutritional factor whose content ranges from 0.3% to 1.09% of dry weight, depending on the species, variety, and environmental conditions [11]. The presence of oxalic acid is relevant for nutritional planning, especially in populations at risk of kidney stones or with mineral absorption concerns.

#### 3.1.1. Superior Value of the Protein and Its Functional Justification

*Lupinus mutabilis* is recognized for an unusually high protein content of around 41–53% of dry weight, far surpassing soybeans’ and five times that of corn and wheat, which challenges the dependence on imported legumes in contexts of food insecurity. To argue that this protein is valuable only for its quantity would be reductive; its quality is equally relevant. The lupin protein matrix is rich in essential amino acids, particularly lysine, which is often deficient in cereals. Evidence suggests that, in food mixtures, *Lupinus mutabilis* can correct the deficient profile of vegetable proteins typical of the regional diet.

A critical question is whether this high protein level is maintained after processing. Herein lies the importance of evaluating not only the initial composition but additionally how industrial processes, particularly debittering, affect nutritional quality. Debittering is essential to making the grain edible, but, according to Salazar et al., it drastically reduces its antioxidant capacity, an effect that is not necessarily negative if toxic components can be eliminated without excessively compromising the beneficial functional compounds.

The functionality of this protein remains relevant even after industrial processing. Debittering, an essential process for removing toxic alkaloids and making the grain suitable for consumption, can alter the nutritional composition; however, studies show that protein levels are not only preserved, they can additionally be concentrated due to the leaching of other soluble components. Although this process reduces the grain’s antioxidant capacity, the balance between food safety and nutritional functionality is usually favorable, especially if toxic compounds can be eliminated without unduly compromising the beneficial bioactive components.

In addition to *Lupinus mutabilis*, *Amaranthus* spp. is also recognized for its valuable protein content, which ranges from 13% to 17% of dry weight. Although lower than that of lupin, amaranth protein is notable for its high nutritional quality and balanced amino acid profile, particularly its richness in lysine and methionine, which are often limited in cereals. Several studies have demonstrated that amaranth proteins possess a high digestibility (up to 90%) and a protein efficiency ratio comparable to that of animal proteins [12]. The inclusion of *Amaranthus* spp. flour in food formulations not only increases the overall protein content but also improves the amino acid balance of cereal-based products, making it a valuable ingredient for enhancing the nutritional quality of both gluten-containing and gluten-free foods. Furthermore, amaranth proteins exhibit functional properties such as water absorption, emulsification, and foaming capacity, which contribute to the technological performance of bakery and extruded products [13].

The potential of *Lupinus mutabilis* as a food ingredient is not limited to its nutritional value. Its flour has outstanding techno-functional properties, such as a water and oil retention capacity and foaming stability, which facilitates its incorporation into baked goods and pastries. Breadmaking trials have shown that the partial replacement of wheat flour with *Lupinus mutabilis* flour, in proportions of up to 20%, 30% and 50% in breads, cookies, and muffins, respectively, not only improves the protein and fiber profile of the final product but additionally maintains a good sensory acceptance. This versatility, combined with its superior nutritional profile and its ability to correct dietary deficiencies, makes *Lupinus mutabilis* a strategic ingredient for the development of functional foods and for dietary diversification in contexts of nutritional transition.

#### 3.1.2. Fat, Fiber, and Carbohydrates: Metabolic Advantages and Comparative Applications in Andean Grains

*Lupinus mutabilis* offers an energetic and functional advantage due to its high fat content, which is around 17%. Beyond their caloric contribution, lupin lipids exhibit an unsaturated profile that may promote cardiovascular health, according to dietary patterns recommended by the FAO/WHO. In addition, its low carbohydrate content (around 7%) and high fiber content (>10%) make it a candidate for foods targeted at diabetic or hyperglycemic populations, justified by its potential to induce a low glycemic rate.

A more detailed analysis reveals that the lipid fraction in *Lupinus mutabilis* is predominantly composed of unsaturated fatty acids, with oleic, linoleic, and linolenic acids accounting for over 70% of its total lipids, which is favorable for cardiovascular health [13]. In contrast, *Amaranthus* spp. contains 6–8% fat, with a significant proportion of squalene and tocopherols, compounds known for their antioxidant and cholesterol-lowering properties [3]. The starch content in *Lupinus mutabilis* is relatively low (6–7%), and it is characterized by a high amylose-to-amylopectin ratio, which contributes to a lower glycemic index and slower digestibility [10]. *Amaranthus* spp., while richer in starch (up to 60% of dry weight), also exhibits a favorable amylose content and the presence of resistant starch, supporting its use in functional foods for glycemic control [12]. Regarding dietary fiber, *Lupinus mutabilis* provides more than 10% total fiber, with a balanced ratio of soluble and insoluble fractions, which aids in satiety, glycemic regulation, and gut health [14]. *Amaranthus* spp. offers 7–10% fiber, mainly insoluble, but also contains mucilaginous polysaccharides that improve water retention and texture in food matrices [15]. These compositional features not only enhance the nutritional value of both grains but also influence their technological performance in food processing.

In *Amaranthus* spp., the richness in protein (13–17%) and fiber (7–10%) is combined with a moderate proportion of fat (6–8%), making it a superior alternative to wheat and rice. This balance supports the argument that pseudocereals can address both protein deficiencies and problems of insufficient fiber in the modern diet.

#### 3.1.3. Bioactive Compounds: Why Are They Relevant and How Are They Affected by Processes?

Talking about bioactive compounds such as polyphenols and flavonoids goes beyond the antioxidant hype; these phytochemicals modulate inflammatory processes, reduce cellular oxidative damage, and may contribute to the prevention of degenerative diseases. Therefore, their presence and preservation after industrial processing take on practical meaning, not just theoretical potential.

A detailed review of the recent literature reveals that both *Lupinus mutabilis* and *Amaranthus* spp. are rich in diverse bioactive compounds, including phenolic acids (such as ferulic, caffeic, and p-coumaric acids), flavonoids (quercetin, rutin), saponins, and, in the case of amaranth, betalains and squalene [6]. These compounds contribute not only to antioxidant activity but also to anti-inflammatory, antihypertensive, and cholesterol-lowering effects. The concentration and profile of these bioactives are highly variable, influenced by genetic factors, environmental conditions, and post-harvest handling.

Technological processes play a decisive role in modulating the content and bioavailability of these compounds. For example, debittering in *Lupinus mutabilis*, while essential for alkaloid removal, can lead to significant losses of water-soluble phenolics and flavonoids, with reductions in total antioxidant capacity exceeding 50% [16]. Conversely, germination and fermentation in *Amaranthus* spp. have been shown to activate endogenous enzymes, resulting in the hydrolysis of bound phenolics and the synthesis of new antioxidant compounds, thereby increasing its total polyphenol content and antioxidant activity [5]. Extrusion and baking, commonly used in food processing, may degrade heat-labile bioactives but can also promote the formation of Maillard reaction products with antioxidant properties [17]. The net effect of these processes depends on parameters such as temperature, duration, and moisture content.

Recent studies highlight that optimizing processing conditions—such as using milder debittering methods, controlled germination, or lower-temperature baking—can help preserve or even enhance the functional potential of these bioactive compounds in the final product [18]. Understanding these mechanisms is crucial for developing functional foods that retain the health-promoting properties of Andean grains.

The evidence indicates that *Lupinus mutabilis* contains polyphenols and shows a high antioxidant activity. However, the debittering process can reduce this activity by more than 50%, posing a dilemma: the treatment that eliminates dangerous alkaloids additionally compromises the food’s functional value. Therefore, it is critical to investigate alternative methods or processing parameters that balance safety and functionality.

In contrast, germination in *Amaranthus* spp. has been shown to be a strategy that not only preserves but additionally increases the grain’s polyphenol concentrations and antioxidant capacity. This fundamental difference between debittering and germination underscores how different processes can antagonistically transform the functional profile of pseudocereals. Sprouting activates hydrolytic enzymes that release phenolic compounds previously bound to the cellular matrix and reduces antinutritional factors, thereby increasing micronutrient bioavailability and proteins’ digestibility. Furthermore, recent research demonstrates that foods made with sprouted *Amaranthus* spp., such as pasta, not only preserve but additionally enhance their antioxidant activity, resulting in greater benefits for the end consumer.

#### 3.1.4. Functional Persistence of Andean Grains in Processed Foods

The integration of Andean grains such as *Lupinus mutabilis* and *Amaranthus* spp. in modern food products has generated considerable interest due to their exceptional protein, fiber, and micronutrient profile. However, the critical question is whether these nutritional and functional benefits are maintained after industrial processing, such as baking or extrusion.

Recent evidence demonstrates that the incorporation of *Amaranthus* spp. flour in baked or extruded products consistently increases the protein and fiber content, surpassing conventional cereals. *Amaranthus* spp., for example, contains between 13% and 18% protein—more than wheat or rice—and offers a complete amino acid profile, particularly rich in lysine, a nutrient lacking in other grains. Its fiber content (up to 8% by dry weight) and high levels of bioactive compounds, such as phenols, squalene, and tocopherols, contribute to improved digestive health and a reduced risk of chronic diseases such as cardiovascular disease and type 2 diabetes. It is important to note that thermal processes (baking, extrusion) can partially degrade some antioxidants, but they can additionally generate new bioactive compounds through Maillard reactions, which could provide similar or even greater benefits.

In the case of *Lupinus mutabilis*, its high protein and oil content, coupled with a low carbohydrate fraction, makes it an attractive ingredient for functional foods, especially in populations at risk of protein malnutrition or metabolic disorders. However, the preservation of its nutritional quality after processing is not guaranteed. For example, the debittering process, essential for removing toxic alkaloids, can reduce its antioxidant capacity, underscoring the need to optimize processing protocols to balance safety and functionality. Similarly, in *Amaranthus* spp., techniques such as soaking, germination, and fermentation have been shown to improve the bioavailability of minerals and reduce antinutritional factors, thus maximizing the healthy impact of the final product.

Despite these promising findings, the translation of the benefits observed in vitro and in their composition into real-life effects on human health is still poorly documented. While community-based interventions in Africa have shown that the integration of *Amaranthus* spp. in the diet can significantly improve children’s growth and micronutrient status, well-designed clinical trials evaluating the bioavailability and physiological effects of these grains after processing in different populations are lacking.

Recent studies on Andean pseudocereals, such as kañihua (*Chenopodium pallidicaule*), reinforce the nutritional value of native species by showing a protein content of up to 19% and amino acid profiles with high concentrations of lysine, threonine, and isoleucine, even surpassing quinoa in some cases. This evidence supports the strategic use of ingredients such as *Amaranthus* spp. and *Lupinus mutabilis*, whose proteins—particularly lupin 11S globulins—offer comparable functional profiles, with a low prolamin content and high biological value. This functional and compositional convergence among Andean species highlights their potential for developing gluten-free food matrices, with applications in both public health and food technology innovation.

#### 3.1.5. Critical Considerations for Industrial-Scale Processing of *Lupinus mutabilis* and *Amaranthus* spp. Flours

The scale-up of *Lupinus mutabilis* and *Amaranthus* spp. flours for industrial applications requires planning at several stages. First, the selection and standardization of raw materials are essential to reduce the variability associated with different cultivars, growing conditions, and post-harvest practices [19]. Developing clear protocols for sourcing and initial processing can help maintain consistency in macronutrient and micronutrient profiles [20].

Processing methods for these flours should be optimized based on evidence from existing pilot and industrial experiences. Debittering processes for lupin, for example, should balance an effective removal of alkaloids with a minimal loss of protein and bioactive compounds [21]. For both grains, heat treatments, extrusion, or fermentation must be validated to minimize nutrient degradation while ensuring food safety [22]. Pilot-scale runs can provide data on yield, functionality, and consistency before moving to larger production volumes [23].

At the formulation stage, it is important to evaluate the performance of Andean flours within diverse food matrices. Studies have shown that blending it with other flours and additives can improve dough’s handling, sensory acceptance, and shelf life [24]. Functional properties, such as water holding capacity, emulsification, and gelation, should be monitored after each scale-up modification [25].

Throughout the scale-up, real-time quality control is necessary. Analytical control points for protein, dietary fiber, antinutritional factors, and bioactive compounds must be implemented, using standardized and validated methodologies [26]. Food safety control—including the rapid detection of microbiological hazards and allergens—should meet the regulatory standards relevant to each market [27].

Consumer studies and sensory evaluation are required to ensure that product quality and acceptance are retained at larger production scales [28]. Such iterative feedback informs adjustments to processing and formulation that help maintain nutritional and functional integrity upon industrialization [29].

### 3.2. Functional and Technological Properties

Flours from *Lupinus mutabilis* and *Amaranthus* spp. go beyond a simple description of their attributes to become a multidimensional forum for debate, where food innovation, sustainability, and real functionality converge—and sometimes even conflict. Therefore, analyzing their rheological properties, digestibility, and the effects of technological processes requires a critical approach that questions both the reported benefits and the existing limitations (Table 2).

First, the water absorption capacity, gelling behavior, and exceptional protein–fiber content of *Lupinus mutabilis* show promise that, in the ideal of food reformulation, could displace traditional components of low nutritional value. However, optimism is tempered when the consequences of processing are recognized: procedures such as debittering, essential for food safety, can reduce its antioxidant capacity by more than 50%. This paradox, between technological necessity and functional loss, reveals the urgency of developing lower-impact processing technologies, because otherwise it leads to a central contradiction: “functional” foods lose precisely what makes them innovative if the processing is not carefully optimized.

This dilemma is not exclusive to *Lupinus mutabilis*. In *Amaranthus* spp., germination and extrusion emerge as interventions that, far from being neutral, reshape the functional and technological profile of the food. The germination of *Amaranthus* spp., for example, can increase the concentration of polyphenols and improve the solubility and digestibility of proteins and starches, favoring the creation of more homogeneous and elastic food matrices. This observation, however, does not inaugurate the “end of the story”; rather, it forces us to think of the process as targeted manipulation: germination could reduce viscosity but increase elasticity, which is a dialectic between functional advantages and rheological changes that requires formulation and reformulation strategies.

It is not enough here to celebrate sensory improvements or nutritional superiority; rather, there is an urgent argument that the increased digestibility achieved through technological processes must be interpreted considering the goal of public health and well-being. Thus, while extrusion enhances digestibility and texture, does this procedure not additionally induce the loss of certain bioactive compounds or generate products with higher glycemic indices? Extrusion, as a high-temperature process, can denature proteins and gelatinize starches, improving their availability; however it can additionally trigger the generation of undesirable compounds or reduce biological functionality if left unchecked.

The use of structural characteristics such as viscosity, pseudoplasticity, and cohesiveness provided by *Amaranthus* spp. in mixtures opens a window for the development of baked goods and pastas adapted to “gluten-free” trends and healthier matrices. But the argument must go beyond techno functionalism: while the formation of a homogeneous matrix favors texture and potentially digestibility, this value must be validated at the clinical and population level, measuring its real effect on the glycemic response and the bioavailability of micronutrients.

Bringing the discussion towards mixtures between Andean pseudocereals such as *Lupinus mutabilis*, *Chenopodium quinoa,* and *Amaranthus* spp. implies recognizing the power of synergy, which becomes a biotechnological and sociocultural argument. By combining legumes and pseudocereals, not only is the protein quality and texture of the final products increased, a regenerative value chain is additionally built that challenges monotonous food models dependent on exogenous inputs. But this argument, to be convincing, requires more than technical descriptions: it must be demonstrated that these synergies are stable on an industrial scale and that they can overcome both organoleptic and commercial barriers.

The debate on the functional and technological properties of *Lupinus mutabilis* and *Amaranthus* spp. cannot be reduced to the sum of its descriptors. It is essential to emphasize that true innovation lies in the ability to maximize its nutritional and technological benefits without sacrificing the integrity of its beneficial compounds, overcoming processing paradoxes and adopting a holistic approach that recognizes the interaction between composition, process, functionality, and public health. Only in this way will these resilient and bioculturally valuable crops fulfill their promise of revolutionizing food’s formulation and functionalization in Latin America and beyond.

### 3.3. Applications in Food Development

The shift in the use of Andean grains from traditional niches to contemporary food matrices, such as breads, gluten-free products, dairy alternatives, and supplements, represents one of the most important and innovative developments in 21st-century food science. Its relevance lies both in its compositional and technological transformation and in its potential implications for food security, public health, and industrial diversification (Table 3).

An important aspect to consider when incorporating amaranth flour into food products is its relatively high fat content (6–8%), which, while nutritionally beneficial due to its unsaturated fatty acid profile and the presence of squalene and tocopherols [3], can increase the susceptibility of amaranth-based products to lipid oxidation during processing and storage. Oxidative stability is a critical quality parameter, as lipid oxidation can negatively affect the sensory attributes, nutritional value, and shelf life of the final products [45]. Although several studies have reported the nutritional and functional benefits of amaranth supplementation [12], systematic research specifically addressing the oxidative stability of these products remains limited. Some evidence suggests that the natural antioxidants present in amaranth, such as tocopherols and polyphenols, may confer a degree of protection against oxidation [29]; however, their effectiveness depends on processing conditions, storage environment, and the presence of pro-oxidant factors. Therefore, future research should focus on evaluating the oxidative stability of amaranth-enriched foods, optimizing processing parameters, and potentially incorporating additional natural antioxidants to ensure product quality and consumer safety. Addressing oxidative stability is essential for the successful industrial application of unconventional ingredients like amaranth flour.

#### 3.3.1. Baking: Innovation for Nutrition and Technological Functionality

Applications in baking, both in conventional and gluten-free products, reveal the disruptive potential of grains such as *Chenopodium quinoa*, *Amaranthus* spp., *Chenopodium pallidicaule*, and *Lupinus mutabilis*. The incorporation of these ingredients at levels even higher than 10–20% significantly increases the protein content (up to 13–16%), fiber, and minerals, dimensions directly associated with a reduction in the glycemic index and the promotion of satiety and metabolic well-being. These effects are not merely additive: they articulate a synergy, since the texture and structure of the crumb (cohesiveness, firmness) often surpass those of traditional gluten-free products based on refined starches, which represents a direct solution to the technological deficiency of conventional gluten-free products.

However, the evidence is critical and guided by sensory and acceptability criteria: it is observed that massive substitution (>20% in bread, >30% in biscuits) begins to negatively impact sensory acceptance, which reveals a technical–nutritional duality in the face of organoleptic barriers. At this point, food science must position itself, as multiple authors point out, at the forefront of biotechnology capable of modulating processes (pretreatments, blends, applied enzymology) aimed not only at maximizing functional benefits but additionally at preserving sensory attributes essential for market success.

Extruded snacks and alternative panettone additionally demonstrate the technological flexibility of these grains, validating the hypothesis that the potential for food expansion lies not only in nutritional capacity but additionally in the ability to adapt to industrial extrusion, baking, and mixing processes, while maintaining or improving sensory acceptability and texture. This is particularly relevant for the functional foods sector, where crunchiness and cohesiveness determine parameters in quality perception.

In addition to the recognition of the nutritional and functional properties of Andean flours such as *Lupinus mutabilis*, studies highlight their economic potential in the development of functional baked goods. The incorporation of lupin flour, combined with amaranth and lactulose in roll-type bread rolls (“Magiya”), not only improved the nutritional profile—with a 36.7% increase in protein and a 22.8% increase in magnesium—but additionally achieved significant economic efficiency, yielding a profit margin of over 6000 rubles per ton produced. This evidence suggests that the use of functional flours of plant origin can be simultaneously viable from technological, nutritional, and commercial perspectives.

Regarding the maximum levels of addition without inhibiting fermentation in baked goods, the current literature indicates that flours from Andean *Lupinus mutabilis* and *Amaranthus* spp. can be incorporated into cereal-based doughs at levels of up to 10–20% for *Amaranthus* spp. and up to 10–15% for *Lupinus mutabilis*, when mixed with wheat flour, without significantly affecting fermentation or loaf volume [46]. Higher inclusion rates may lead to reduced gas retention and a suboptimal texture due to gluten dilution and increased fiber and non-gluten protein content.

#### 3.3.2. Dairy Products and Beverages as a Challenge for Plant-Based Innovation

The addition of Andean cereals to dairy matrices, such as yogurt or alternative cheeses, demonstrates a dual effect: on the one hand, nutritional enrichment (increased protein, fiber, fat, and bioactive compounds); on the other, challenges associated with sensory acceptability, which significantly decreases when concentrations exceed 3–5%. Simply adding superfoods does not guarantee sensory success; research into selective fermentation, starter selection, and encapsulation technologies are needed to mitigate undesirable changes in flavor, texture, or color.

The mixture of *Amaranthus* spp. and *Lupinus mutabilis* has made it possible to increase protein and antioxidant content, raising sensory acceptability to scores of 7–9 out of 10. This result underscores the importance of strategic formulations and a thorough understanding of ingredient–process interactions, an aspect strongly emphasized in the recent literature.

#### 3.3.3. Other Functional Foods: Public Health Perspective

The substitution of animal fat with Andean pseudocereals in meat products and the creation of protein-functional supplements from Andean grain blends increase protein and fiber while decreasing total fat content, thus providing real and scientifically validated alternatives to address problems such as obesity, cardiovascular disease, and malnutrition due to excess or deficiency. Fortification with *Amaranthus* spp. and *Lupinus mutabilis* demonstrates a clear increase in iron and phenolic compounds, with direct implications in combating anemia and promoting the antioxidant health of the population.

However, the fundamental challenge here, evidenced by the variability in sensory acceptability, is the same one that runs through the entire field: science must operate not only at the innovative frontier of nutrients and functional compounds but additionally in recognition of consumer expectations and habits, the cultural context, and the regulatory framework.

### 3.4. Sensory Evaluation and Consumer Acceptance

Sensory evaluation constitutes the critical link that connects the technological and functional developments of innovative ingredients, such as Andean grains, with the reality of mass consumption and market viability (Table 4). Its scientific basis lies in the ability of sensory methods to translate the physicochemical and structural variations produced by the substitution or addition of *Amaranthus* spp. and *Lupinus mutabilis* and other Andean pseudocereals and legumes into objective and validated judgments on preference, acceptance, and purchasing potential.

**Table 4 foods-14-02059-t004:** Sensory evaluation and acceptance factors in products based on Andean grains.

Product/Food Matrix	Sensory Method Used	Number of Consumers/Panelists	Acceptance and Preference (Scale, % Acceptance, Highlighted Attributes)	Factors Influencing Sensory Perception	Reference
Gluten-free bread (*Chenopodium quinoa*, *Amaranthus* spp., *Chenopodium pallidicaule*, *Lupinus mutabilis*)	Hedonic scale with 9 points, CATA, GPA, ANOVA, MFA	100–250 consumers (varies by study)	Optimal acceptability with ≤20% substitution; >30% decreases acceptance; key attributes: texture, color, flavor, fluffy crumb	Substitution level, texture, color, bitter taste of *Chenopodium quinoa*, type of preference, presence of phenolic compounds	[47]
Panettone (*Chenopodium quinoa*, *Amaranthus* spp.)	CATA, 9-point hedonic scale, preference ranking, Friedman	80 consumers	Acceptability like commercial with ≤15% substitution; preferred attributes: fluffy, sweet, moist, vanilla scent; preference for PE and PB samples	Type of preference, proportion of *Chenopodium quinoa*/*Amaranthus* spp., sensory attributes (smell, texture, sweetness)	[48]
Gluten-free cookies (*Chenopodium pallidicaule*, *Chenopodium quinoa*, *Amaranthus* spp.)	Hedonic scale with 9 points, Sorting, CA, ANOVA	102 consumers	Greater acceptance with 20–30% *Chenopodium Pallidicaule*; attributes: crisp texture, darker color, pleasant flavor; acceptability >7/9 in better formulations	*Chenopodium* ratio *pallidicaule*, texture (hardness, crispness), color, starch, and protein content	[49]
Vegetable burger (*Chenopodium quinoa*, *Lupinus mutabilis*, *Amaranthus* spp.)	CATA, 5-point hedonic scale, CA	132 consumers	High acceptability (mostly “like it a lot” or “like it”); attributes: easy to cut, soft, legume flavor, healthy	Proportion of ingredients, texture, flavor, color, perception of healthiness	[50]
Vegetable yogurt (*Chenopodium quinoa*, *Lupinus mutabilis*)	Hedonic scale 9 points, JAR, ANOVA	50–100 consumers	Acceptability decreases >3% *Chenopodium Quinoa*; attributes: flavor, creamy texture, color; optimal acceptability with ≤3% addition	Level of addition, texture, aftertaste, sweetness	[51]
Fortified biscuits (*Amaranthus* spp. and *Chenopodium pallidicaule*)	9-point hedonic scale, Sorting, CA	102 consumers	Optimal acceptability with ≤30% *Amaranthus* spp./*Chenopodium pallidicaule*; attributes: crunchy texture, pleasant flavor, dark color	Proportion of fortifier, texture, color, flavor	[40]
Bars/supplements (*Chenopodium quinoa*, *Amaranthus* spp., *Chenopodium pallidicaule*)	Hedonic scale 7–9 points, ANOVA	50 consumers	Good acceptability (>6/9); attributes: flavor, texture, color, healthy perception	Proportion of ingredients, texture, sweetness, aftertaste	[52]

#### 3.4.1. Methodological Rigor and Diversity of Sensory Tools

The recurrent use of the hedonic scale as a reference method for quantifying liking highlights the need for an international comparability and reproducibility of results. This technique, when applied to a sufficient range of consumers (50 to 250, depending on the product and study), allows not only the identification of overall acceptability but additionally the determination of technologically critical substitution thresholds, as revealed in the case of gluten-free bread and extruded snacks, where optimal acceptance is restricted to substitutions of ≤20%, and >30% implies significant drops in acceptance.

The CATA (Check-All-That-Apply) methodology and multivariate statistical analyses (MFA, GPA, ANOVA, Friedman) reinforce the robustness of designs, allowing one to explore not only the degree of liking but additionally the relative relevance and interaction of specific sensory attributes (color, texture, flavor, aroma), generating perceptual maps whose interpretation can guide reformulations. Complementarily, the JAR (Just -About-Right) and Pivot technique profiles address the ideality of individual attributes, a crucial dimension in fortified or functional matrices.

#### 3.4.2. Necessary Balance Between Innovation and Acceptability

Modern food science must recognize that innovative advances in formulations are only meaningful if they pass the sensory test. In baked goods—such as biscuits, pastries, and snacks—evidence demonstrates that there is a threshold range in which the substitution or addition of Andean grains maximizes nutritional and functional benefits (e.g., increased protein, fiber, minerals, and bioavailable compounds) without compromising sensory acceptability. However, beyond certain levels (e.g., more than 30% in Chenopodium quinoa, or over 3% in plant-based yogurts), sensory limitations tend to appear, often due to bitterness, intense coloration, gritty textures, or undesirable changes in cohesion and elasticity.

In addition to the evidence on *Chenopodium quinoa* and *Amaranthus* spp., several studies have evaluated the sensory impact of incorporating *Lupinus mutabilis* flour into bakery and snack products. The partial substitution of wheat flour with *Lupinus mutabilis* (typically up to 10–15%) has been shown to significantly increase protein and fiber content while maintaining good sensory acceptance, especially in breads and cookies [36]. However, higher inclusion levels may introduce a characteristic “leguminous” flavor and denser texture, which can affect consumer preferences. The optimization of formulations and processing—such as combining *Lupinus mutabilis* with other pseudocereals or using flavor-masking strategies—has proven effective in enhancing both nutritional value and acceptability [33]. These findings underscore the importance of *Lupinus mutabilis* as a functional ingredient that can be successfully integrated into innovative food products without compromising sensory quality when used at appropriate levels.

At the level of consumer preference, the fact that panettone made with *Chenopodium quinoa* and *Amaranthus* spp. (up to a 15% substitution) maintains acceptability comparable to commercial products—or that vegetable burgers incorporating Andean ingredients achieve high sensory scores (“I such as it a lot”)—suggests that the challenge lies not in mere incorporation but in the specific optimization of ingredient matrices, processes, and proportions.

#### 3.4.3. Determining Factors in Sensory Perception

The analysis of results demonstrates that sensory perception is the product of a multivariate and interdisciplinary interaction:Level of substitution/addition and type of processing: A key factor that modulates the manifestation of desirable attributes. High substitutions can accentuate phenolic compounds or saponins, responsible for undesirable flavors and dark colors.Texture: Characteristics such as crunchiness, fluffiness, cohesiveness, or hardness are determining factors for preference and usually respond to both the ingredient matrix and the technological treatment (extrusion, baking, fermentation), directly influencing acceptance.Flavor and aroma: A “leguminous flavor”, sweetness, nutty notes, or the bitterness typical of some Andean compounds act as limiting parameters and are subject to adjustments through pretreatments or mixtures.Color and appearance: For a large proportion of consumers, the “dark color” of *Chenopodium pallidicaule* or *Amaranthus* spp. represents a negative factor if it deviates from the traditional visual expectation associated with the reference food.Perception of healthiness: In functional products, the perception of “natural” ingredients and knowledge of the benefits can compensate for slight disadvantages in taste or texture.

The rigor of sensory analysis as a scientific discipline ensures that the transfer from the laboratory to industry is carried out under empirical validation and that advances based on Andean grains have real viability in the commercial, nutritional, and public spheres.

### 3.5. Impact on Health and Functional Potential

Recent scientific literature converges in pointing out that Andean grains constitute food matrices with an outstanding functional potential and a positive impact on human health, supported by three main axes: antioxidant activity, a reduction in metabolic risk factors and the contribution of fiber and high-quality proteins.

#### 3.5.1. Antioxidant Activity

Various studies have shown that Andean grains possess a high antioxidant capacity, attributed to their content of phenolic compounds, flavonoids, betalains, and other phytochemicals. For example, *Chenopodium quinoa* contains between 30.3 and 59.7 mg FA/100 g of total phenolic acids, including caffeic, ferulic, and p-coumaric acids. *Amaranthus* spp. and *Chenopodium pallidicaule* additionally stand out for their phenolic and antioxidant profiles.

The antioxidant activity of these grains has been reported not only in crude extracts but is additionally maintained—or even enhanced—after technological processes such as germination, extrusion, or enzymatic hydrolysis. For example, the germination of *Chenopodium quinoa* and *Amaranthus* spp. increases the accumulation of bioactive compounds and boosts their antioxidant capacity, while extrusion can improve their antioxidant bioavailability, although it may additionally cause losses depending on the processing conditions.

In the case of *Lupinus mutabilis* and pajuro, the enzymatic hydrolysis of their proteins generates multifunctional peptides with potent antioxidant activity, capable of protecting cells from oxidative stress. These peptides have demonstrated, in vitro, the ability to scavenge free radicals and prevent cell damage induced by oxidative agents.

#### 3.5.2. Reduction in Metabolic Risk Factors

The consumption of Andean grains is associated with a reduction in metabolic risk factors such as hypertension, hyperglycemia, dyslipidemia, and obesity, due to the presence of bioactive peptides, dietary fiber, and phenolic compounds.

In the case of *Lupinus mutabilis*, its high protein and dietary fiber content, along with a low glycemic carbohydrate fraction, make it particularly effective in reducing the postprandial glycemic response and improving lipid profiles. Clinical and preclinical studies have demonstrated that the inclusion of *Lupinus mutabilis* flour in the diet can contribute to lower blood glucose levels, reduced total cholesterol, and improved satiety, supporting its use in the management of diabetes and cardiovascular risk [53]. Moreover, bioactive peptides derived from *Lupinus mutabilis* proteins have shown inhibitory activity against enzymes involved in hypertension and glycemic regulation, further highlighting its metabolic health benefits [19]. These effects complement those observed with other Andean grains and reinforce the value of *Lupinus mutabilis* as a key ingredient in functional foods aimed at reducing metabolic risk factors.

Protein hydrolysates from Chenopodium quinoa, *Amaranthus* spp., *Lupinus mutabilis*, and Erythrina edulis Triana have been shown to inhibit angiotensin-converting enzyme I (ACE), suggesting antihypertensive potential. Furthermore, these hydrolysates exhibit inhibitory activity against α-amylase, α-glucosidase, and DPP-IV—key enzymes involved in glycemic regulation—indicating antidiabetic potential.

The dietary fiber content in *Chenopodium quinoa*, *Amaranthus* spp., and *Chenopodium pallidicaule* ranges from 7% to 16% on average, with higher values reported in *Chenopodium pallidicaule* and *Amaranthus* spp. This fiber contributes to reducing the glycemic index, improving satiety, and modulating the gut microbiota—factors associated with the prevention of obesity and other metabolic disorders.

#### 3.5.3. Fiber and Protein Intake

Andean grains stand out for their high-quality protein content, featuring complete amino acid profiles and a high digestibility. For example, *Chenopodium quinoa* and *Amaranthus* spp. contain between 13% and 18% protein, while *Lupinus mutabilis* and pajuro can exceed 40–50%. Furthermore, the in vitro digestibility of Chenopodium quinoa and *Lupinus mutabilis* proteins can reach values above 80%.

Dietary fiber is another relevant component, with values ranging from 7% to 16% in *Chenopodium quinoa*, *Amaranthus* spp., and *Chenopodium pallidicaule* and up to 18% in *Lupinus mutabilis*. This fiber, along with resistant starches and other polysaccharides, contributes to digestive health, a reduction in cholesterol, and the modulationof postprandial glucose. It is worth noting that recent studies have begun to characterize the amino acid profile of products made with *Lupinus mutabilis* flour. For example, Paz-Yépez et al. developed a vegan dressing containing 9% lupin flour, which reached a protein content of 5.68%, highlighting essential amino acids such as threonine (0.93 g/100 g), leucine (0.63 g/100 g), and histidine (0.62 g/100 g), along with high levels of glutamic acid (2.21 g/100 g). This finding demonstrates that lupin not only provides a high quantitative protein fraction but additionally retains its nutritional quality after technological processes such as emulsification, reinforcing its functional value in plant-based food formulations.

## 4. Conclusions

The evidence compiled in this review demonstrates that both *Lupinus mutabilis* and *Amaranthus* spp. possess a remarkable density of nutrients and functional compounds, with significant variations attributable to genetic, geographical, and technological factors. The incorporation of *Lupinus mutabilis* enables a substantial increase in protein content compared to cereals such as wheat and maize, while *Amaranthus* spp. stands out for its antioxidant capacity, providing up to 180 mg of polyphenols per 100 g. Applications of these grains in modern food matrices have been shown to elevate protein and fiber content by 10% to 40%, maintaining an optimal sensory acceptance at substitution levels below 20–30%. Although certain technological processes may reduce some bioactive compounds, nutritional functionality can often be preserved or even enhanced through these processes’ optimization. Unlike previous studies that typically address nutritional properties or technological applications separately, this review offers a holistic and current synthesis of the nutritional composition, technological processing, retention of bioactives, and health-promoting properties for both species. By integrating the recent literature, this review provides a unique comparative perspective on how processing strategies affect the retention of nutrients and functional compounds, identifies challenges and opportunities for the development of innovative food products, and fills important knowledge gaps regarding the valorization and sustainable use of *Lupinus mutabilis* and *Amaranthus* spp. in functional foods and modern food systems.

## Figures and Tables

**Figure 1 foods-14-02059-f001:**
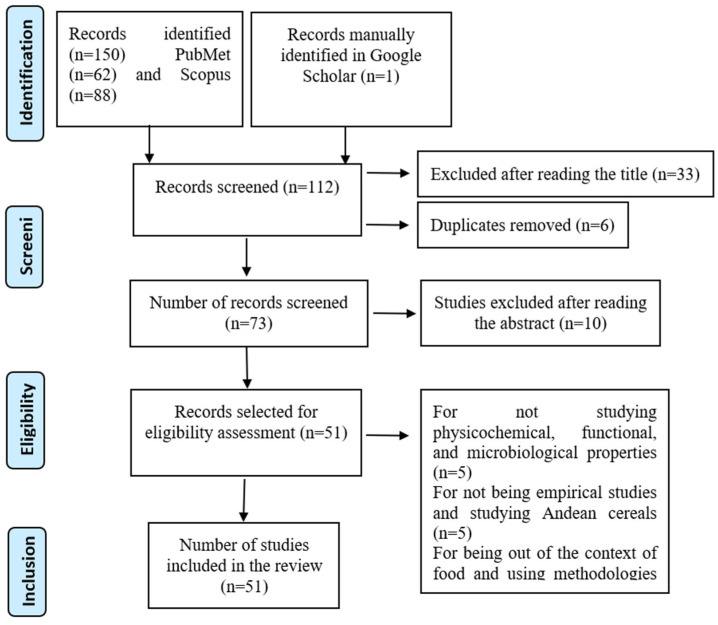
PRISMA flow diagram of the selection of literature for the systematic review.

**Table 1 foods-14-02059-t001:** Nutritional composition and bioactive compounds of Andean grains.

Species	Proximal Composition Parameters	Bioactive Compounds	Relevant Observations	References
*Lupinus mutabilis*	Protein: 52.8%, Fat: ~17%, Carbohydrates: 6.9%, Fiber: >10%, and Moisture: 5.94–18.87%	Antioxidants: Bittering reduces antioxidant capacity by 52.9%, and spray-drying reduces antioxidant capacity by an additional 8%. Phenols and flavonoids are present.	*Lupinus mutabilis* is notable for its extremely high protein and fat content. The bitterness reduces its antioxidant capacity. It meets the requirements for “high fiber.” Variability is attributed to geography and variety.	[4]
*Lupinus mutabilis*	Protein: 41–45%, Fat: 16–18%, and Fiber: 9–13%	Polyphenols, antioxidant capacity.	Technological processes (bittering, drying) affect the antioxidant capacity and phenol content.	[5]
*Lupinus mutabilis* (mixture with quinoa and sweet potato)	It does not report individual values, but the mix increases protein and fiber in extruded products.	Does not report individual values.	Being extruded with *Lupinus mutabilis*, quinoa, and sweet potato improves the protein and fiber profile.	[6]
*Amaranthus* spp.	Protein: 13–17%, Fat: 6–8%, and Fiber: 7–10%	Total polyphenols: 80–180 mg FA/100 g.	High genetic variability. *Amaranthus* spp. is a relevant source of antioxidants.	[7]
*Amaranthus* spp.	Protein: 13–16%, Fat: 6–8%, Dietary fiber, minerals	Polyphenols, flavonoids, saponins, antioxidants.	Germination increases bioactive compounds and antioxidant capacity.	[8]
*Amaranthus* spp. (germinated)	It does not report individual numerical values, but germination improves digestibility and nutritional profile.	Increase in polyphenol and antioxidant capacity after germination.	Use of *Amaranthus* spp. germinated in paste improves nutritional and functional potential.	[9]
*Amaranthus* spp. (in baking)	Increasing protein and fiber in baked goods with *Amaranthus* spp.	Does not report individual values.	Improves the nutritional profile of baked goods.	[10]

**Table 2 foods-14-02059-t002:** Functional and technological properties of and processing effects on *Lupinus mutabilis* flours and mixtures, and *Amaranthus* spp.

Species/Mixture	Functional and Technological Properties	Rheological and Textural Properties	Digestibility of Starches and Proteins	Effect of Technological Processes (Germination, Extrusion, Drying, etc.)	Relevant Observations	Reference
*Lupinus mutabilis*	High water and oil absorption capacity; gelling properties; high fiber (>10%), protein (>52%), and fat (~17%) content.	Viscoelastic gel-like behavior in doughs; high starch gelatinization temperature (68.4–81.5 °C) due to the presence of non-starch compounds; medium-sized particles.	Starch: low content (6.9%), but high proportion of amylose; high-quality proteins, rich in lysine.	Bittering reduces antioxidant capacity by 52.9%, and spray-drying reduces antioxidant capacity by an additional 8%; it maintains the integrity of starch and proteins after milling.	Meets the requirements for “high fiber” status; variability attributed to geographic area and variety; suitable for gluten-free products.	[30]
*Lupinus mutabilis*	Functional capacity affected by processes; significant reduction in polyphenols and antioxidants after debittering and drying.	It does not report numerical values, but it is mentioned that processing affects texture and functionality.	Direct digestibility is not reported, but it is inferred that the reduction in polyphenols may affect bioavailability.	Bittering and drying reduce the antioxidant capacity and phenol content.	Technological processes can compromise the functional value if they are not optimized.	[31]
Mix: *Lupinus mutabilis* + quinoa + sweet potato (extruded)	Increased protein and fiber content in extruded products; improved nutritional and functional profile; products with greater water absorption capacity.	Extrusion produces products with a crisp texture and good cohesiveness and expansion; it improves sensory acceptability.	Extrusion can increase the digestibility of proteins and starches by denaturation and gelatinization.	Extrusion improves texture, digestibility, and nutritional value; individual values are not reported.	Positive synergy between Andean ingredients; suitable for healthy snacks.	[32]
*Amaranthus* spp.	High water absorption capacity; significant source of polyphenols (80–180 mg FA/100 g); genetic variability in functional properties.	Flours with fine particles; good gel formation; suitable for baking and extruded products.	It does not report direct digestibility but highlights the presence of high-quality proteins and starch with good functionality.	Genetic and environmental variability affect functional properties; specific processes are not assessed.	*Amaranthus* spp. is a significant source of antioxidants and fiber; suitable for functional products.	[33]
*Amaranthus* spp. (germinated)	Sprouting increases polyphenols, flavonoids, and antioxidant capacity; it improves water absorption and protein solubility.	Germination reduces pastes’ viscosity but improves the elasticity and cohesiveness of doughs; it facilitates the formation of more homogeneous matrices.	Sprouting increases protein digestibility and mineral bioavailability; it improves starch’s digestibility.	Germination: Increased bioactive compounds and antioxidant capacity; reduction in antinutritional factors; improved technological functionality.	Germination is an optimal process to enhance the functional and technological value of *Amaranthus* spp.	[34]
*Amaranthus* spp. (germinated)	Use of *Amaranthus* spp. germinated in pasta improves the nutritional and functional profile; increases polyphenols and antioxidants in the final product.	Pasta with *Amaranthus* spp. germinated have better texture (greater firmness and elasticity) and sensory acceptability.	Germination improves the digestibility of proteins and starch in the produced pasta.	Germination prior to pasta production increases bioactive compounds and functionality; it improves the texture and digestibility of the final product.	Germination is key to the development of functional foods from *Amaranthus* spp.	[35]
*Amaranthus* spp. (in baking)	Increased protein and fiber content in breads; improved nutritional profile; good water absorption capacity in doughs.	Breads with *Amaranthus* spp. They have a good crumb, greater volume, and acceptable texture; they improve the elasticity and cohesiveness of the dough.	It does not report direct digestibility, but baking can improve the bioavailability of nutrients.	Baking: Possible partial loss of antioxidants but improves products’ texture and acceptability.	*Amaranthus* spp. It is useful for enriching baked goods and improving their functionality.	[36]
Mixtures: wheat, quinoa, *Amaranthus* spp.	*Amaranthus* spp. improves technofunctional properties (water absorption, swelling, apparent density); mixtures exhibit variability in color and microstructure.	*Amaranthus* spp. Pure: higher apparent viscosity and flow resistance; mixtures with wheat and quinoa: changes in pseudoplasticity and cohesiveness; microstructure: *Amaranthus* spp. provides a more homogeneous and finer matrix.	It does not report direct digestibility, but the fine and cohesive structure favors digestion in baked products.	Blending and fine grinding improve cohesiveness and texture; variations in color and microstructure depend on proportions.	*Amaranthus* spp. is key to improving functionality and texture in bread and pastry mixes.	[37]
Mixtures: quinoa, *Amaranthus* spp., *Lupinus mutabilis* (in paste)	Mixtures improve functional and textural properties of pastes, increasing firmness, cohesiveness, and viscosity.	Pastas with a higher proportion of *Lupinus mutabilis* and *Amaranthus* spp.: greater firmness and cohesiveness; final viscosity and retrogradation increase with *Amaranthus* spp.	It does not report direct digestibility, but the combination of flours improves the texture and potential digestibility.	Blends allow for fine-tuning the texture and functionality of pasta; cooking improves structure and acceptability.	Positive synergy between pseudocereals and Andean legumes in aqueous matrices.	[38]

**Table 3 foods-14-02059-t003:** Applications of Andean grains in food development.

Application/Product	Andean Grain/Mixture	Level of Substitution/Addition	Nutritional Composition	Reference
Baking: gluten-free bread	*Chenopodium quinoa*, *Amaranthus* spp., *Chenopodium pallidicaule*, *Lupinus mutabilis*	10–50% addition to starch/potato/corn	Increased protein (up to 13–16%), fiber, minerals, and bread volume (with 10–20% *Chenopodium quinoa*/*Amaranthus* spp.); improved texture (cohesiveness, firmness); optimal acceptability with ≤20%.	[39]
Baking: cookies	*Chenopodium pallidicaule*, *Chenopodium quinoa*, *Amaranthus* spp.	10–40% replacement	Increased protein (up to 12–15%), fiber, and minerals; optimal sensory acceptability with 20–30% Chenopodium pallidicaule; darker color.	[40]
Baking: extruded snacks	*Chenopodium quinoa*, *Lupinus mutabilis*, sweet potato	20–40% mix	Improved protein (up to 14–18%), fiber, expansion, and crispness; high sensory acceptability.	[35]
Baking: panettone	*Chenopodium quinoa*, *Amaranthus* spp.	10–20% replacement	Increased protein, fiber, and minerals; sensory acceptability like a commercial product with ≤15%.	[41]
Dairy products: alternative cheese	*Chenopodium quinoa*, *Amaranthus* spp.	10–20% replacement	Increased protein and fiber; acceptable texture; moderate sensory acceptability	[42]
Others: supplements/bars	*Chenopodium quinoa*, *Amaranthus* spp., *Chenopodium pallidicaule*	10–40% mix	Increased protein (up to 15–18%), fiber, and minerals; good sensory acceptability	[43]
Others: fortified foods	*Chenopodium quinoa*, *Lupinus mutabilis*	10–20% addition	Increased iron, protein, fiber, and phenolic compounds; variable sensory acceptability.	[44]

## Data Availability

The original contributions presented in this study are included in the article. Further inquiries can be directed to the corresponding author.

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
