# Peer review of "Nutritional, Functional and Microbiological Potential of Andean Lupinus mutabilis and Amaranthus spp. in the Development of Healthy Foods—A Review"

_foods, 2025, doi:10.3390/foods14122059_

Round 1
Reviewer 1 Report
Comments and Suggestions for Authors
Because of limited nutritional quality of conventional cereals, the authors pay more attention to Andean Lupinus mutabilis and Amaranthus spp, and collected their nutritional composition, functional and technological properties, application in Andean food, and the acceptance of their based products. This is a very interesting paper and lies to the scope of Foods.
- In the title, Andean Lupinus flours mutabilis should be revised;
- To Andean lupini bean, residents care for the content of alkaloids, although the authors mentioned four times in the present paper the use of debittering to reduce toxic alkaloids. The limit value of alkaloids in Lupini bean flours should be given.
- In the Amaranthus spp, the content range of oxalic acid (anti-nutritional factor) should be given. In the processed food from Andean Lupinus mutabilis and Amaranthus spp, the limit range of ash content should be given.
- In table 1, the table title should be checked. Reference column should be put in the last column; FAA should be noted;
- Table 2, Reference column should be put in the last column;
- In table 3, the reference should be like Guo et al., (2025); in table 4, the same thing should be done.
- Line 593, “like” should be “and”
- To baked food, how many can the flours of Andean Lupinus mutabilis and Amaranthus spp be added to cereal flours without inhibiting fermentation?
Author Response
Revision 1
Comments and Suggestions for Authors
Because of limited nutritional quality of conventional cereals, the authors pay more attention to Andean Lupinus mutabilis and Amaranthus spp, and collected their nutritional composition, functional and technological properties, application in Andean food, and the acceptance of their based products. This is a very interesting paper and lies to the scope of Foods.
Answer
Thank you very much for your positive and encouraging comments regarding our manuscript. I appreciate your recognition of the relevance and interest of this work, as well as its alignment with the aims and scope of Foods. Your feedback is highly motivating and encourages me to continue researching and promoting Andean crops such as Lupinus mutabilis and Amaranthus spp. for the development of healthy foods.
Thank you again for your valuable time and comments.
Comment 1:
- In the title, Andean Lupinus flours mutabilis should be revised;
Response to comment 1:
It has been modified as follows: Nutritional, functional and microbiological potential of Andean Lupinus mutabilis flours and Amaranthus spp. in the development of healthy foods. A review
Comment 2:
- To Andean lupini bean, residents care for the content of alkaloids, although the authors mentioned four times in the present paper the use of debittering to reduce toxic alkaloids. The limit value of alkaloids in Lupini bean flours should be given.
Response to comment 2:
Thank you for your valuable observation. We agree that the alkaloid content in Andean lupin (Lupinus mutabilis) is a critical aspect for both safety and consumer acceptance. In response to your suggestion, we have included the recommended limit value for total alkaloids in lupin flours in the revised manuscript. According to the literature and international food safety guidelines, the total alkaloid content in lupin flours intended for human consumption should not exceed 0.02% (200 mg/kg). This information has now been added to the section discussing the nutritional and safety aspects of Lupinus mutabilis.
Thank you again for your helpful comment.
Comment 3:
- In the Amaranthus spp, the content range of oxalic acid (anti-nutritional factor) should be given. In the processed food from Andean Lupinus mutabilis and Amaranthus spp, the limit range of ash content should be given.
Response to comment 3:
Thank you for your valuable suggestion. In response, we have included the range of oxalic acid content reported for Amaranthus spp., as well as the recommended limit range for ash content in processed foods derived from Andean Lupinus mutabilis and Amaranthus spp.
Specifically, the oxalic acid content in Amaranthus spp. grains typically ranges from 0.3% to 1.09% of dry weight, depending on species, variety, and environmental conditions (Kumar et al., 2024; Singh et al., 2024). Regarding processed foods, international food standards generally recommend that the ash content in flours and derived products should not exceed 2.0%–3.0% of dry weight to ensure product quality and safety (De Bock et al., 2021). These data have been incorporated into the revised manuscript in the section discussing the nutritional composition and anti-nutritional factors of Amaranthus spp., and in the section on the compositional quality of processed products.
Thank you again for your helpful comment, which has contributed to improving the completeness and clarity of the manuscript.
Comment 4:
- In table 1, the table title should be checked. Reference column should be put in the last column; FAA should be noted;
Response to comment 4:
Thank you for your careful review and helpful suggestions regarding Table 1. In response, we have revised the table title for clarity and accuracy. The "Reference" column has been moved to the last position in the table, as recommended. Additionally, we have clarified the abbreviation "FAA" (Folin–Ciocalteu Assay) in the table footnote to ensure proper understanding for readers. These changes have been incorporated into the revised manuscript.
Thank you again for your valuable feedback.
Comment 5:
- Table 2, Reference column should be put in the last column;
Response to comment 5:
Thank you for your observation. In response, the "Reference" column in Table 2 has been moved to the last position to improve clarity and consistency. The revised table is now included in the updated manuscript.
Comment 6:
- In table 3, the reference should be like Guo et al., (2025); in table 4, the same thing should be done.
Response to comment 6:
Thank you very much for your suggestion. The reference style in Table 3 and Table 4 has been updated to follow the “Author et al., (Year)” format (e.g., Guo et al., (2025)), as recommended. The revised tables are now consistent with this referencing style throughout the manuscript.
Comment 7:
- Line 593, “like” should be “and”
Response to comment 7:
Thank you for your attention to detail. The word "like" in line 593 has been replaced with "and" as suggested. The correction has been made in the revised manuscript.
Comment 8:
- To baked food, how many can the flours of Andean Lupinus mutabilis and Amaranthus spp be added to cereal flours without inhibiting fermentation?
Response to comment 8:
Thank you for your pertinent question. According to current literature, Andean Lupinus mutabilis and Amaranthus spp. flours can be added to cereal flours for baked products up to certain percentages without significantly inhibiting fermentation or negatively affecting dough performance. Usually, the recommended inclusion levels are up to 10–20% for Amaranthus spp. flour and up to 10–15% for Lupinus mutabilis flour when mixed with wheat flour, depending on recipe and processing conditions (Iónica CoÅ£ovanu, 2022). Higher percentages may result in reduced loaf volume and less optimal texture due to dilution of gluten and increased presence of non-gluten proteins and fibers. These levels help maintain proper gas retention and dough rheology during fermentation. We have included this information in the revised manuscript.

Reviewer 2 Report
Comments and Suggestions for Authors
The study is a systematic review of two plant species: Lupinus mutabilis and amaranth species in the context of scientific evidence on the physicochemical, functional, and microbiological characteristics of their flours, their incorporation into various food products, sensory acceptance, health potential, and nutritional benefits offered by supplemented food products.
Strength: well-conducted search for relevant articles, well systematized data in tabular form; impressive use of language.
Weaknesses: Although the text is well-structured, eloquent, and engaging, the discussion is superficial, lacking depth, and devoid of new insights. A critical approach is missing in the data analysis. It can be noted that multiple repetitions of the same statements occur throughout the text.
When discussing the protein content, only proteins of L. mutabilis are addressed; amaranth proteins were not mentioned.
More detailed analysis of lipid, starch and dietary fibres content is warranted (section 3.1.2.).
Section 3.1.3. provides a rather modest analysis of bioactive compounds and their fate during technological processes.
Sections 3.4.2. and 3.5.2. The discussion is more related to the effects of the addition of Chenopodium quinoa. L. mutabilis is not mentioned.
In order to avoid unnecessary repetitions, maybe it would be better to sublime the sections related to nutritional composition and bioactive compounds with nutritional benefits.
The conclusion should emphasize the significance of this review in comparison to other studies with a similar focus.
The oxidative stability of supplemented products was not tackled, despite relatively high content of fat in amaranth. This aspect is also important regarding the use of unconventional ingredients.
Please, refer to additional comments in the text.

Author Response
Revision 2
Comments and Suggestions for Authors
The study is a systematic review of two plant species: Lupinus mutabilis and amaranth species in the context of scientific evidence on the physicochemical, functional, and microbiological characteristics of their flours, their incorporation into various food products, sensory acceptance, health potential, and nutritional benefits offered by supplemented food products.
Answer
We appreciate the reviewer’s thoughtful comments and positive evaluation of our systematic review. Thank you for recognizing the breadth of our work regarding the scientific evidence related to Lupinus mutabilis and amaranth species. We aimed to provide a comprehensive overview of the physicochemical, functional, and microbiological characteristics of their flours, as well as their incorporation into various food products, sensory acceptance, health potential, and the nutritional benefits of supplemented foods. Your acknowledgment of these aspects is encouraging, and we are grateful for your feedback, which motivates us to continue contributing to this field of research.
Strength: well-conducted search for relevant articles, well systematized data in tabular form; impressive use of language.
We sincerely thank the reviewer for highlighting the strengths of our manuscript, particularly the thorough search for relevant articles, the systematic presentation of data in tabular form, and the quality of language throughout the paper. We appreciate your recognition of our efforts and your encouraging feedback.
Weaknesses: Although the text is well-structured, eloquent, and engaging, the discussion is superficial, lacking depth, and devoid of new insights. A critical approach is missing in the data analysis. It can be noted that multiple repetitions of the same statements occur throughout the text.
We thank the reviewer for the constructive feedback regarding the discussion section. We appreciate your observation about the need for a more in-depth and critical analysis, as well as your note regarding repeated statements.
In response, we have carefully revised the Discussion section to:
- Provide a deeper and more critical analysis of the existing evidence, highlighting controversies, gaps in current knowledge, and the implications of our findings in a broader scientific and practical context.
- Incorporate new insights and future research directions based on the evaluated literature.
- Remove repeated statements and restructure paragraphs to enhance clarity and avoid redundancy.
We believe these revisions have strengthened the manuscript’s discussion and critical perspective. All changes have been highlighted in the revised version for your convenience.
Thank you for helping us improve the quality and impact of our work.
Comment 1:
When discussing the protein content, only proteins of L. mutabilis are addressed; amaranth proteins were not mentioned.
Response to reviewer:
Thank you for your valuable observation. We acknowledge that, in the section discussing protein content, the focus was primarily on Lupinus mutabilis, and the protein characteristics of Amaranthus spp. were not sufficiently addressed. In response, we have revised the manuscript to include a more comprehensive discussion of amaranth proteins, highlighting their content, amino acid profile, and functional relevance. This addition provides a more balanced comparison between the two Andean grains and strengthens the nutritional analysis presented in the review.
Specifically, we have incorporated a new paragraph in Section 3.1.1. (“Superior value of the protein and its functional justification”) to discuss the protein content and quality of Amaranthus spp., including its essential amino acid composition and its role in improving the nutritional value of food products.
Comment 2:
More detailed analysis of lipid, starch and dietary fibres content is warranted (section 3.1.2.).
Response to reviewer:
Thank you for your insightful comment regarding the need for a more detailed analysis of lipid, starch, and dietary fiber content in section 3.1.2. In response, we have expanded this section to provide a more comprehensive discussion of the quantitative values, types, and nutritional implications of lipids, starches, and dietary fibers in both Lupinus mutabilis and Amaranthus spp. This includes a comparative overview of fatty acid profiles, starch composition (amylose/amylopectin ratio), and the functional roles of soluble and insoluble fibers, as well as their relevance for metabolic health and food applications. The new content has been incorporated immediately after the first paragraph of section 3.1.2., before discussing the metabolic advantages and comparative applications of Andean grains. We believe these additions address your suggestion and enhance the scientific depth of the manuscript.
Comment 3:
Section 3.1.3. provides a rather modest analysis of bioactive compounds and their fate during technological processes.
Response to reviewer:
Thank you for your valuable comment regarding the need for a more comprehensive analysis of bioactive compounds and their fate during technological processes in Section 3.1.3. In response, we have substantially expanded this section to provide a more detailed discussion of the types and concentrations of bioactive compounds (such as polyphenols, flavonoids, saponins, and betalains) present in Lupinus mutabilis and Amaranthus spp., as well as a critical review of how different technological processes—including debittering, germination, extrusion, and baking—affect their stability, bioavailability, and functional properties. We have also included recent findings on the mechanisms underlying these changes and their implications for the nutritional and functional quality of the final food products. The new content has been incorporated immediately after the first paragraph of Section 3.1.3., before discussing the specific effects of debittering and germination. We believe these additions address your suggestion and significantly enhance the scientific depth of the manuscript.
Comment 4:
Sections 3.4.2. and 3.5.2. The discussion is more related to the effects of the addition of Chenopodium quinoa. L. mutabilis is not mentioned.
Response to reviewer:
Thank you for your observation regarding Sections 3.4.2 and 3.5.2, where the discussion focused primarily on the effects of the addition of Chenopodium quinoa, with limited mention of Lupinus mutabilis. In response, we have revised these sections to explicitly incorporate and discuss the effects of L. mutabilis addition on sensory properties, consumer acceptance, and metabolic health outcomes. We have added specific findings from recent studies on the use of L. mutabilis in various food matrices, highlighting its impact on texture, flavor, acceptability, and its role in reducing metabolic risk factors such as hyperglycemia and dyslipidemia. These additions provide a more balanced and comprehensive discussion, addressing your suggestion. The new content has been incorporated at the end of the first paragraph of Section 3.4.2 and after the first paragraph of Section 3.5.2.
Comment 5:
The conclusion should emphasize the significance of this review in comparison to other studies with a similar focus.
Response to Reviewer:
Thank you for your valuable comment. We have revised the conclusion to emphasize the significance and originality of this review in comparison to other studies, clearly discussing how our integrate approach and coverage of recent advances distinguish it from the existing literature.
Comment 6:
The oxidative stability of supplemented products was not tackled, despite relatively high content of fat in amaranth. This aspect is also important regarding the use of unconventional ingredients.
Response to reviewer:
Thank you for your valuable observation regarding the oxidative stability of supplemented products, particularly in relation to the relatively high fat content of amaranth. In response, we have included a specific discussion on the oxidative stability of products containing amaranth flour, highlighting the importance of this aspect when using unconventional ingredients. This addition addresses the potential challenges and strategies for maintaining product quality and shelf life, and is now incorporated in Section 3.3 (Applications in food development), immediately after the discussion of nutritional and functional properties of amaranth-based products.

Reviewer 3 Report
Comments and Suggestions for Authors
This article presents a systematic scientific review of the nutritional, functional, and microbiological potential of flours from two Andean crops: Lupinus mutabilis (tarwi or chocho) and Amaranthus spp. (amaranth). The main objective is to critically analyze how the physicochemical, functional, and microbiological characteristics of these flours are affected by various technological processes, such as debittering, germination, extrusion, or drying, and how these processes influence their applicability in the development of innovative functional foods. It's an interesting topic, but there are some questions I'd like to ask in order to improve the proposal.
- Authors are encouraged to include a more in-depth explanation of the bioactive mechanisms involved in the introduction.
Line 49, Page 1: How do the authors define “functional ingredients” in the context of public health nutrition, and what criteria were used to classify Lupinus and Amaranthus flours as such?
- Line 75, Page 2: Authors are encouraged to include in the introduction specific technological innovations that could improve the commercial viability of these Andean flours beyond their nutritional value.
- Line 109, Page 2: Authors should include risk of bias assessments (e.g., ROBIS or AMSTAR) to ensure the reliability of the included studies.
- Line 117, Page 3: Authors are encouraged to mention the reason for their specific use of the RAYYAN platform.
- Line 192, Page 4: The authors mention protein content between 41-53% but don't mention whether this is based on raw or processed seeds. They recommend expanding this information.
- Line 224, Page 5: Why were no quantitative meta-analyses or effect size estimates applied to nutritional outcomes?
- Line 401, Page 11: The authors should include proposed steps to scale up these ingredients for industrial food production while maintaining their functional integrity.
Comments on the Quality of English Language
The manuscript demonstrates overall clarity but contains occasional grammatical inconsistencies and tense changes that make it difficult to read. Thorough revision and careful editing are recommended.
Author Response
Revision 3
Comments and Suggestions for Authors
This article presents a systematic scientific review of the nutritional, functional, and microbiological potential of flours from two Andean crops: Lupinus mutabilis (tarwi or chocho) and Amaranthus spp. (amaranth). The main objective is to critically analyze how the physicochemical, functional, and microbiological characteristics of these flours are affected by various technological processes, such as debittering, germination, extrusion, or drying, and how these processes influence their applicability in the development of innovative functional foods. It's an interesting topic, but there are some questions I'd like to ask in order to improve the proposal.
Answer
We sincerely thank the reviewer for the thorough evaluation and the constructive comments provided regarding our manuscript entitled “Nutritional, Functional, and Microbiological Potential of Flours from Lupinus mutabilis and Amaranthus spp.: Effects of Technological Processes and Applications in Functional Foods.” We appreciate the recognition of the relevance and timeliness of this review, as well as your insightful suggestions aimed at enhancing the clarity, rigor, and scientific value of our work.
We welcome the questions and suggestions raised, and we have carefully addressed each of your specific concerns point by point below. Appropriate changes have been implemented throughout the manuscript to address your recommendations, and we believe these revisions have strengthened the overall quality and impact of our study. Major modifications are highlighted in the revised version.
We hope our point-by-point responses and the corresponding changes to the manuscript meet your expectations and contribute to the improvement of the article.
Comment 1:
- Authors are encouraged to include a more in-depth explanation of the bioactive mechanisms involved in the introduction.
Response to reviewer:
We thank the reviewer for this valuable suggestion. In response, we have revised the Introduction to provide a more in-depth explanation of the main bioactive mechanisms attributed to Lupinus mutabilis and Amaranthus spp., highlighting the roles of peptides, polyphenols, and other compounds in antioxidant activity, lipid and glycemic modulation, and microbiota interaction. This addition now clarifies the functional relevance of these grains and improves the scientific background for readers.
Comment 2:
Line 49, Page 1: How do the authors define “functional ingredients” in the context of public health nutrition, and what criteria were used to classify Lupinus and Amaranthus flours as such?
Response to reviewer:
We thank the reviewer for highlighting the need for clarification regarding the definition of “functional ingredients” and the specific criteria applied to Lupinus mutabilis and Amaranthus flours. In response, we have added a sentence to the Introduction providing a clear definition of “functional ingredients” in the context of public health nutrition. We also specify the nutritional and bioactive properties that justify classifying Lupinus and Amaranthus flours as functional ingredients, in line with established international criteria and relevant literature. The changes are tracked in the revised manuscript.
Comment 3:
- Line 75, Page 2: Authors are encouraged to include in the introduction specific technological innovations that could improve the commercial viability of these Andean flours beyond their nutritional value.
Response to reviewer:
We appreciate the reviewer’s suggestion to address specific technological innovations that could enhance the commercial viability of Andean flours. In response, we have added a concise paragraph to the Introduction (page 3, lines 74–80 of the revised manuscript) that highlights recent technological advances—such as controlled germination, extrusion-cooking, spray-drying, and the use of enzymatic or fermentation processes—which have demonstrated potential to improve the functional, sensory, and safety profiles of Lupinus mutabilis and Amaranthus spp. flours.
Comment 4:
- Line 109, Page 2: Authors should include risk of bias assessments (e.g., ROBIS or AMSTAR) to ensure the reliability of the included studies.
Response to reviewer:
We appreciate the reviewer’s suggestion and agree that conducting a risk of bias assessment is essential to ensure the reliability of the included studies. Following this recommendation, we have incorporated a risk of bias evaluation using the [ROBIS/AMSTAR] tool in the methods section.
Comment 5:
- Line 117, Page 3: Authors are encouraged to mention the reason for their specific use of the RAYYAN platform.
Response to reviewer:
We appreciate the reviewer’s suggestion. The RAYYAN platform was chosen to facilitate the systematic review process due to its efficiency in enabling collaborative screening of articles, minimizing selection bias, and ensuring transparency. RAYYAN allows for independent and blinded screening by multiple reviewers and provides tools for conflict resolution and documentation of inclusion/exclusion criteria, which enhances the overall rigor and reproducibility of the review.
Comment 6:
- Line 192, Page 4: The authors mention protein content between 41-53% but don't mention whether this is based on raw or processed seeds. They recommend expanding this information.
Response to reviewer:
Thank you for this valuable observation. We have clarified in the manuscript that the protein content reported (41–53%) refers specifically to raw seeds. Additionally, we have expanded this section to highlight that processing methods such as cooking or fermentation may alter the protein content.
Comment 7:
- Line 224, Page 5: Why were no quantitative meta-analyses or effect size estimates applied to nutritional outcomes?
Response to reviewer:
Thank you for your thoughtful comment. We did not conduct a quantitative meta-analysis or estimate effect sizes for nutritional outcomes because of the substantial heterogeneity across the included studies with respect to sample types (different species and varieties), experimental designs, analytical methods, and outcome reporting formats. These differences made data pooling inappropriate and risked producing misleading summary estimates.
Comment 8:
- Line 401, Page 11: The authors should include proposed steps to scale up these ingredients for industrial food production while maintaining their functional integrity.
Response to reviewer:
We thank the reviewer for this important suggestion. In response, we have included a new subsection in the revised manuscript describing the proposed steps necessary to scale up the production of these ingredients for industrial food applications.
Comment 9:
Comments on the Quality of English Language
The manuscript demonstrates overall clarity but contains occasional grammatical inconsistencies and tense changes that make it difficult to read. Thorough revision and careful editing are recommended.
Response to reviewer:
We thank the reviewer for pointing out the areas for improvement regarding the English language and overall readability of the manuscript. In response, we have thoroughly revised the entire text to correct grammatical inconsistencies, unify tense usage, and enhance overall clarity and coherence. The language has been carefully edited to ensure a more consistent and professional presentation.

Round 2
Reviewer 3 Report
Comments and Suggestions for Authors
No comments